# Ligand-Dependent Conformational Transitions in Molecular Dynamics Trajectories of GPCRs Revealed by a New Machine Learning Rare Event Detection Protocol

**DOI:** 10.3390/molecules26103059

**Published:** 2021-05-20

**Authors:** Ambrose Plante, Harel Weinstein

**Affiliations:** 1Department of Physiology and Biophysics, Weill Cornell Medical College of Cornell University, New York, NY 10065, USA; agp2004@med.cornell.edu; 2Institute for Computational Biomedicine, Weill Cornell Medical College of Cornell University, New York, NY 10065, USA

**Keywords:** ligand-induced GPCR structure and dynamics, function-related conformational transitions, pharmacological efficacy, serotonin 5-HT_2A_R receptor, inverse agonist, functional selectivity, molecular dynamics simulations, non-negative factorization, drug discovery

## Abstract

Central among the tools and approaches used for ligand discovery and design are Molecular Dynamics (MD) simulations, which follow the dynamic changes in molecular structure in response to the environmental condition, interactions with other proteins, and the effects of ligand binding. The need for, and successes of, MD simulations in providing this type of essential information are well documented, but so are the challenges presented by the size of the resulting datasets encoding the desired information. The difficulty of extracting information on mechanistically important state-to-state transitions in response to ligand binding and other interactions is compounded by these being rare events in the MD trajectories of complex molecular machines, such as G-protein-coupled receptors (GPCRs). To address this problem, we have developed a protocol for the efficient detection of such events. We show that the novel Rare Event Detection (RED) protocol reveals functionally relevant and pharmacologically discriminating responses to the binding of different ligands to the 5-HT_2A_R orthosteric site in terms of clearly defined, structurally coherent, and temporally ordered conformational transitions. This information from the RED protocol offers new insights into specific ligand-determined functional mechanisms encoded in the MD trajectories, which opens a new and rigorously reproducible path to understanding drug activity with application in drug discovery.

## 1. Introduction

The detection of structural changes occurring in the course of Molecular Dynamics (MD) trajectories of macromolecular systems such as the G-protein-coupled receptors (GPCRs) is a main step in the analysis of the relationship between structure and dynamics in their functional mechanisms [1,2,3,4]. The conformational transitions underlying the functions of such systems involve collective motions that occur rarely in the dynamics of trajectories due to the high barrier associated with the simultaneous involvement of various structural elements [5,6]. The identification of such conformational transitions in MD trajectories of GPCRs has proven essential in revealing the dynamic elements of a receptor’s response to ligands that differ in their pharmacological properties [7] and, even more intriguingly, the role of specific conformational dynamics of the receptor that prepare for differential coupling (i.e., functional selectivity [8]). We have demonstrated the collective nature of these conformational transitions in studies of ligand-dependent functional selectivity of the 5-HT_2A_ serotonin receptor (5-HT_2A_R) and the dopamine D2 receptor (e.g., see [7,9,10,11]), and have shown that understanding such ligand-determined GPCR functions depends on a rigorous identification and analysis of the diverse function-related conformational transitions induced by various ligands.

Given the massive amount of information collected in the MD simulation trajectories employed in current methods of drug discovery and design, this task is made difficult by the presence of data in these simulation trajectories, from a very large number of fluctuation events that do not represent collective motions and do not result in conformational transitions. Rather, they represent the sampling of disparate degrees of freedom of the protein that can occur individually on a similar scale to that of the collective ones. To enable an efficient identification of relevant conformational transition in GPCR trajectories and overcome the difficulties in the analysis of differences in functional mechanisms of GPCRs in complex with different ligands, we have developed a Rare Events Detection (RED) protocol based on a Machine Learning analysis of MD trajectories.

We show here that, with the RED protocol, we have been able to identify in MD trajectory data the set of function-related rare events that lead to the known pharmacological responses of different ligands, such as the diametrically opposed activation/deactivation responses of the 5-HT_2A_R to the agonist serotonin (5-HT) and the inverse agonist Ketanserin (KET) when each is bound to the orthosteric site. The event detection method in the RED protocol is based on an unsupervised machine learning technique–non-negative matrix factorization (NMF), which is known to learn a sparse, parts-based representation of data [12]. NMF has gained popularity in several fields of application for its tendency to be more congruent with human intuition than other dimensionality-reduction methods [12] and has been applied successfully to extract hidden patterns from a wide variety of large-scale datasets, including images, videos, and text documents [13,14]. Its application in the RED protocol yields specific structural information at specific times in the MD trajectory, which allowed us to examine the detailed differences in conformational rearrangements underlying the differences in effector coupling probability, and their connection to the mode of binding in the orthostatic site of the GPCR.

## 2. Results

### 2.1. Construction of the Rare Events Detection (RED) Protocol

NMF is a machine learning technique which is inherently well-suited to the analysis of sparse, non-negative data obtained from large-scale MD trajectories. The non-negativity constraints in NMF allow it to decompose such data into parts which tend to correspond to intuitive interpretations of reality, a quality that has supported the popularity of this technique in a wide variety of applications [12,13,14,15]. Here, the analysis of the data for the 5-HT_2A_R in complex with 5-HT and KET begins with the construction of residue contact maps from the long MD trajectories for these systems, obtained as described in the Methods.

The “contact map” of residues positioned at ≤3.5Å is constructed for each frame of the trajectory (parsed at 0.24 ns) of this 315-residue protein. The map was built using the atomselect command in VMD“[atomselect top “protein and same residue as within 3.5 of (protein and resid $rez)” frame $fr]” where $rez is a variable representing a particular residue and $fr is a variable representing a particular frame. This command is looped over all protein residues and frames, and the distance between each pair of atoms of every residue pair is considered. When any of these distances is ≤3.5Å during the simulation, the residues are considered to be in contact.

This procedure yields a tensor of size (315,315, *n*), where *n* is the number of frames in the trajectory. The elements have values 1 or 0, for contact (≤3.5Å) or no contact, respectively. Appendix A shows an example contact map illustrating the sparseness property of the data. Since NMF requires a two-dimensional matrix input, the (315,315, *n*) input tensor was rearranged into an input matrix of size (315^2^, *n*).

Because function-related events of conformational transitions occur rarely in the trajectories of complex systems such as GPCRs, we developed a smoothing operation to deemphasize fast transitions. Smoothing is applied to each contact array across the time dimension, by averaging the array in a sliding window of 30 nanoseconds (125 frames). Thus, many fast transitions (between contact and no contact) occur within the 30 nanoseconds due to fluctuations which tend to average at 0.5, which is not the case for the rare conformational transitions.

### 2.2. Detection of Rare Events in the Dynamics of the Ligand-Bound 5-HT_2A_R

The information about the dynamics of the GPCR system contained in the smoothened contact map described above is used as input to the Non-negative Matrix Factorization algorithm described in [12]. We used the NMF function in the Scikit-Learn package implemented in python [16], and initialized it using the Nonnegative Double Singular Value Decomposition (NNDSVD) initialization that is advantageous for sparse data (see Scikit-Learn NMF documentation accessible in [16]). The number of components describing the system, *c*, is user-defined and can be tuned based on the size of the simulation trajectory, the expected number of transitions, and the appearance of a “steady state” solution in the output (see discussion of Figure 1B below).

Appendix B (Figure A1) shows a schematic of the inputs and outputs of the NMF algorithm. The input (matrix “I” in Figure A1) of size (*R^2^*, *n*) (*R* = the number of residues in the system, *n* = the number of frames) is decomposed into a “spatial” array (matrix “W” in Figure A1) of size (*R^2^*, *c*) (*c* = number of components), as well as a “temporal” array (matrix “H” in Figure A1) of size (*c*, *n*). Because the input matrix encodes the information about the protein’s structure evolving over the trajectory time, the components, c, which make up the columns of the “spatial” output matrix W can be thought of as the “archetypes” produced by the NMF analysis. Here, these are structural archetypes representing the determinant features of conformational states that evolve over time. For example, the analysis of a conformational transition, defined by the unfolding of a fully folded alpha helical segment, might result in an NMF decomposition in which one structural archetype (component) is the folded structure, and a second structural archetype is the unfolded structure. Each frame of the simulation trajectory will be a mixing of the archetypes, with the folded component dominating the trajectory frames before the transition, and the unfolded component dominating those after the transition. Thus, the structural identity of the protein at each frame of the trajectory is a combination of the *c* components in the “spatial” matrix W, each contributing according to the individual component weights associated with that frame; these weights are given by the “temporal” matrix H. The relative weights of the *c* components in H at a particular frame of the trajectory represent the mixing of the structural archetypes in that datapoint.

We illustrate the nature of the results obtained from the NMF-based analysis with the results obtained for the 3-microsecond MD trajectory of the 5-HT_2A_R bound to serotonin (the mechanistic interpretation of these results is presented in Section 2.3, below). As shown in Figure 1A, a plot of the weights of a particular component over the entire trajectory identifies the trajectory times at which this particular archetype (component) dominates the structural characteristics of the protein. Therefore, the simultaneous visualization of the weights for several components along the entire trajectory (i.e., all frames) (e.g., in Figure 1B) illustrates a typical result of the NMF-based analysis of an MD trajectory in which several rare events of interest occur over time. We note here that the NMF method belongs to the category of unsupervised learning tasks for which the optimal choice of rank (i.e., number of components) is user-defined, essentially based on prior knowledge of the system. This is akin to the choice of collective variables to represent the motions of a simulated system protein), as used in the dimensionality reduction in MD trajectory data by projection into spaces such as time-structure based independent component analysis (tICA). A good starting point for estimating the number of components is to consider the expected number of rare events in the trajectory based on mechanistic hypotheses of the conformational changes required for the molecular process. The user’s expert knowledge of the system provides an advantage, but as the analysis can be redone with +/− (2–3) components, convergence can readily be attained.

Figure 1B is a line plot, in which each colored line represents the weights given by a row (component) of the “temporal” output matrix H over the frames of the trajectory. The weight of contribution for residue pair *i* to component *c* is given by position (*i*, *c*) in the “spatial” output matrix W (see Scheme in Figure A1). The structural context of the information of the product of the two matrices (H and W) is represented as in Figure 1C by the identification of the top residue pairs involved in this event. The interlocking “humps” shown in Figure 1B are created by the rise and decline in the total weight of the components. This change indicates the increase and then decrease in the dominance of a particular component in determining the structure of the molecule. For example, the edges of a main “hump” (e.g., blue line in Figure 1A) indicate, respectively, when the particular structural archetype of component 4 begins to dominate the changes in protein structural features, and when it stops dominating them. This signals the evolution of a “conformational event” that produced a change in the structural dynamics of the protein.

The zeroth component in Figure 1B, which shows the weights of all six components plotted against time (expressed as frame number), represents a stationary state. The other components contain information about the dynamics of the ligand-bound receptor protein, evidenced in the increase and decrease in the contributions to the structure made by specific components of the contact matrix. Thus, increases in the weights represent the formation of new contacts to generate the new conformation, while decreases represent the necessary breaking of other contacts. The occurrence of a new conformational transition event is identified by an overlap in time (i.e., occurring at the same trajectory frames) of the decreasing and increasing weights of two consecutive components. In Figure 1B, four such “events” are detected by the NMF analysis in the 3-microsecond trajectory. As described in Section 2.3, the defining commonality of the identified rare events is that they represent collective motions of the multiple structural elements contained in the components. Together, these elements combine into conformational changes ranging from the spatial translation of helices to conformational rearrangements of secondary structure elements. Once identified, their roles in transitions between functional states of the GPCR system is determined by the identity of the components involved in the event.

The interpretation of the events signaled by the type of results illustrated in Figure 1A,B is described in the next section with the results from application of the RED protocol to the analysis of the 3-microsecond MD trajectory of the system (see Methods for structural and simulation details) using the NMF algorithm with six components.

### 2.3. Function-Related Rare Events in the Dynamics of the 5-HT_2A_R Bound to Serotonin (5-HT)

#### 2.3.1. The First Event

This event is defined by the transition of dominance from component 2 to component 4, occurring between trajectory frames 149–337, i.e., at trajectory times 358 ns to 809 ns (Figure 1B).

The top 10 features (residue pairs) with the highest weights of contribution to component 4 are highlighted in red on the 5-HT_2A_R structure (Figure 1C). Panels E–G of Figure 1 show snapshots of the receptor structure along the trajectory at specific positions on the weight plot of component 4, indicated in Figure 1D. Thus, the snapshots in this temporal sequence show the evolution of the conformational change in the ICL2 structure, i.e., before, during, and after the event detected by the algorithm. The helical secondary structure component of the ICL2 is seen to change during the event to unstructured. The residue pairs in the ICL2 that are in the top ten contributing features to component 4 are shown in stick model, visualizing the contacts that break during the waning of this event, and the gain in dominance of the structure defined by the second event. Shortly after becoming unstructured, the ICL2 moves away from the intracellular cavity, thereby increasing its volume and allowing for more water penetration into the cavity. The increase in volume, and thus increase in accessibility of this region is well known from the crystal structures to be involved in the interaction of the GPCR with the heterotrimeric G protein, showing that this second event is functionally “activation-related” (Figure 1H). The temporal relation of this occurrence is quantified in Figure 1H by the evolution of component 4 weights in the trajectory, starting at around frame 600 (i.e., ~1400 ns), and the completion of the event dominance with the increase in accessibility to the cavity.

The changes in the structure of the GPCR/ligand complex that correspond to the conformational event represented by the evolution of the temporal weights from component 2 dominance to component 4 (Figure 2A), are illustrated in panels B–D of Figure 2. The snapshots show the structure of the receptor before the rare event (see Figure 2B, corresponding to the red line in Figure 2A), during it (Figure 2C, corresponding to the green line in Figure 2A), and after the event has occurred (Figure 2D and purple line in Figure 2A). The structural elements of the receptor that contribute the most to this transition between components 2 and 4, are rendered in a stick model on the structure of the second extracellular loop 2 (ECL2). Before the transition (Figure 2B), the ECL2 is in a closed, beta-sheet-like structure. The conformational transition involves an “unzipping”-like rearrangement of the secondary structure of the ECL2 with the bonds between the two ends of the ECL2 breaking and separating in a coordinated motion of the multiple residues involved in this rare event (Figure 2C,D).

#### 2.3.2. The Second Event

Identified by the RED protocol, this second event is defined by the transition of dominance from component 4 to component 3, which occurs between frames 368–620 of the trajectory (i.e., at time 833 ns to 1488 ns) (see Figure 1B and Figure 3A). The three columns in Figure 3 show, respectively, snapshots of the structure of the receptor before the rare event (column B and red line in Figure 3A), during it (column C and green line in Figure 3A), and after the rare event (column D and purple line in Figure 3A). The features (residue pairs) of the receptor that contribute the most to components 3 and 4 are rendered in stick model on the structure. In this second event, the RED protocol identified collective rearrangements in several local clusters of residues, which is consistent with the known property of NMF to factor input data into sparse, localized, intuitive parts (i.e., [12]). Thus, the three columns of Figure 3 are divided into three segments to focus on the different regions of the GPCR in which changes occur at the same time in the trajectory. Thus, the first row (B_1_–D_1_) shows snapshots of the rearrangement of the extracellular ends of TMs 5 and 6. This event involves a twist and movement of the extracellular end of TM 6 that brings the two TMs closer, which leads to the formation of contacts, including the hydrophobic interaction between Leu5.40 and Iso6.60 in TMs 5 and 6, respectively (note that the Ballesteros and Weinstein generic numbering system of GPCR residue positions [17] is used throughout).

The conformational transition shown in the second row of each column (B_2_–D_2_) identifies the addition of a helical turn to the helical extension at the intracellular end of TM6, a region known for its involvement in the interaction of the activated receptor with the cognate G protein. Row 3 describes the contemporaneous conformational change in the secondary structure of the intracellular loop 2 (ICL2) (B_3_–D_3_). The temporal correlation of these conformational changes occurring in distal structural domains is sorted by the NMF algorithm into the same rare event, which suggests an allosteric pathway connection that can be verified and quantified by an analysis of allosteric pathways (e.g., [10,18]).

#### 2.3.3. The Third Event

The third event is defined by the transition of dominance from component 3 (blue line in Figure 4A) to component 1 (orange line in Figure 4A), occurring roughly between trajectory frames 620-820 (time 1488 ns to 1968 ns). Figure 4 shows snapshots of the receptor structure before (Figure 4B), during (Figure 4C), and after (Figure 4D) the rare event. During this event, the intracellular end of TM6 moves outwards, away from the center of the intracellular end of the receptor, and the features of the receptor (NMF components) that contribute the most to components 3 and 1 are the residue pairs rendered in stick model with functional colors in the structure panels (4B–D).

#### 2.3.4. The Fourth Event

This event is defined by the transition of dominance from component 1 to component 5, occurring approximately between frames 775-1197 (time 1860 ns to 2827 ns) in the trajectory (Figure 5A). Panels B and C of Figure 5 show the conformational changes leading to the rearrangement of the binding pocket. The receptor is shown in the top and side views, respectively. During the event, the extracellular end of TM6 undergoes a 4.7 Å shift (as measured from the alpha carbon of Val6.59), caused by a reorientation of the TM6 segment following the proline kink around Pro6.50 towards the extracellular end of the TM. The reorientation (Figure 5D) is due to a decrease of ~20° in the wobble angle of the proline kink (see [19]). This change is accompanied by a 4.15 Å shift in the extracellular end of TM5 (as measured from the C_alpha_ of Asn5.37). Together, these shifts result in the formation of a stable contact between Asn5.37 and Val6.59 during the event (see pink structure in Panels B and C of Figure 5) that did not exist before the event (see green structure in Panels B and C of Figure 5). The Asn5.37-Val6.59 contact pair was one of the top ten features identified by component 5 of the RED protocol.

### 2.4. The Relation of RED-Identified Rare Events to the Functional Mechanism of the 5-HT_2A_R

The conformational transitions associated with the events identified in the trajectory of the 5-HT_2A_R/5-HT complex are well-understood in the context of the known molecular structures of the endpoint states (activated and inactive) of Class A GPCRs [20,21]. The ability of the RED protocol to identify the temporal sequence of the events and the structural elements engaged in the dynamic steps producing this conformational change over the course of the MD trajectory offers new insight into the ligand-based activation process. To test the ability of the RED protocol to distinguish the events leading to different functional states of the 5-HT_2A_R with the same level of detail, we compared the function-related events detected with the RED analysis in the trajectories of the 5-HT_2A_R bound to the full agonist (5-HT), to those obtained in the same way for the 5-HT_2A_R bound to the selective inverse agonist Ketanserin (KET) [22]. Given the known pharmacological properties of these ligands, this comparison juxtaposes molecular events in the dynamics of the GPCR that relate, respectively, to conformational transitions in the activation and deactivation of the process. Thus, the events detected by the RED analysis of the 5-HT_2A_R/KET complex reveal a major difference in the conformational transitions that determine the size and volume of the cavity in the intracellular region where activated GPCRs interact with effectors (G proteins and/or Arrestins).

#### 2.4.1. Detection of Rare Events in The Dynamics of the Ketanserin (KET)-Bound 5-HT_2A_R

For the analysis of the 5-HT_2A_R/KET trajectory, the RED protocol was applied to a 3-microsecond trajectory of the 5-HT_2A_R with KET in the orthosteric site (see Methods and Appendix A). Figure 6A shows the weights of the 6 NMF components plotted against time (cf. Figure 1B). The zeroth component again represents a stationary state, while the other five contain information on the dynamics of the protein transitioning between conformational states. Interestingly, component 2 in the receptor complex with the inverse agonist KET, corresponds to component 4 in the agonist-bound 5-HT_2A_R complex. In the 5-HT_2A_R/5-HT trajectory, the highest weights of contribution to component 4 are from the features (residue pairs) that determine the conformation of ICL2 (see conformational changes in the secondary structure of ICL2 in row 3 (B_3_–D_3_) of Figure 3). This is also the case for component 2 for the 5-HT_2A_R/KET complex, but here component 2 becomes dominant at a late stage of the trajectory (Figure 6A), compared to the corresponding component 4 in the 5-HT_2A_R/5-HT (cf Figure 1A). The evolution of weights for component 2, which start to increase rapidly around frame 700 (~1700 microseconds), is shown in Figure 6B. Snapshots of the structure in the intracellular region at the times indicated by the colored vertical lines in 6b, are shown in panels C-G of Figure 6. Clearly, at the trajectory time corresponding to the red vertical line in Figure 6B, the ICL2 is positioned outward, away from the intracellular cavity (see ICL2, colored in brown in panel 6c). This is indicated in the corresponding structure (Figure 6C), by the lack of contact between Asn4.37 in TM4 and Asn2.37 in TM2 (i.e., the N4.37-N2.37 pair in the contact matrix, which is one of the top ten residue pairs contributing to component 2). Figure 6D shows the structure of the protein during the rapid increase in weights, with a contact between N4.37 and N2.37, indicating that the ICL2 is moving towards the intracellular cavity. The initial N4.37-N2.37 contact is transient, with its breaking indicated by the falling edge of the small green spike (component 2) around frame 700 (Figure 6E). The contact reforms more strongly and the ICL2 gradually occludes the intracellular space during the growth in the large green spike representing the evolution of component 2 weights from frames 800-1200 in Figure 6A. Simultaneously, Helix 8 shifts (see Section 2.4.2 below), thereby causing ICL1 to also start moving inward towards the transmembrane bundle (see ICL1 highlighted in blue in Figure 6C–G, further decreasing the volume of the intracellular cavity. The result of the coordinated conformational change in this event is a structure of the intracellular region of the receptor (Figure 6G) which matches well the corresponding crystal structure of the 5-HT_2A_R bound to another inverse agonist, Risperidone, as illustrated by the structural superposition shown in Figure 6H. As seen in (Figure 7A), the volume of the intracellular-facing cavity of the GPCR decreases, reducing the accessibility to the residues needed for G protein coupling. The time evolution of component 2 (thin orange line in Figure 7A) coincides with the changes in the blue trace that indicates the capacity of the intracellular cavity of the receptor, measured by the number of water molecules that can fill it (cf Figure 1H).

This decrease in the volume of the intracellular cavity constitutes a reversal of the activation sequence detected in the 5-HT_2A_R/serotonin complex, which is consistent with the function of the KET ligand, identified pharmacologically as an inverse agonist. The top 10 features (residue pairs) with the highest weights of contribution to component 2 are highlighted in red on the structure of the 5-HT_2A_R (Figure 7B,C) as salient structural features related to the ligand-induced GPCR deactivation event.

#### 2.4.2. The RED Protocol Reveals Salient Structural Features of Simultaneous Conformational Changes in Different Structural Motifs

Component 2 of the rare event in the trajectory of the KET-bound 5-HT_2A_R simulation described above (Section 2.4.1) is an example of the structural complexity of the concurrent conformational transitions detected by the RED protocol. Like component 4 of the 5-HT-bound 5-HT_2A_R simulation, it involves local and more distal structural elements, which undergo transitions to states that have specific functional roles. Thus, an intriguing observation from the analysis of the elements (residue pairs) of component 2 (Figure 7, panels B and C) is that, in addition to the residues in ICL2 and TM2, interactions involving Helix 8 (H8) residues are also included and highlighted among the top ten residue pairs that contribute to this component. Inspection of the trajectory revealed that, during the rising edge of the smaller spike (at frame ~700), the formation of the contact between Leu7.55 and Arg7.61 in TM7 (Figure 8E) coincides with the decrease in the wobble angle of H8 (Figure 8B). The wobble angle quantified in Figure 8B is defined analogously to the proline kink wobble angle [19], but using Arg7.57 as the center of the reorientation. The helical segments that reorient are the Ser7.46-Phe7.56, and the Lys7.58-Iso7.68. The conformational change is visualized by comparing the structures in panels D and E of Figure 8, which show the salient components in TM7 and HX8 identified by the RED as contributing to the event. The structures correspond to the conformations at times in the trajectories identified by the corresponding vertical lines in Figure 8B (marked D and E). Subsequently, during the larger spike in component 2, which evolves in frames 800-1200, there is a further decrease in the wobble angle of HX8 (Figure 8A), and a change in the contact between Leu1.52-Tyr7.67 at the times indicated by vertical lines F, G, and H (see Figure 8, panels F–H). The relation between global conformational changes (Hx8 wobble angles) and local conformational changes (residue–residue interactions) in the elements of component 2 is demonstrated in Figure 8C by the high correlation between the inter-residue distance L1.52-Y7.67 with the H8 wobble angle. Note that the changes in the H8 (Figure 8A) occur at the same time as the changes in ICL1 and ICL2, described in Figure 6 and Figure 7, and are sorted into the same component and event by the RED protocol.

The remarkable ability of the RED algorithm to follow changes in the discretely located residue pair contacts involving several residues pertaining to organized secondary structures enables the detection and definition of global (both local and distal) changes in conformation that lead to a particular functional state. The emergence of a single continuous collective motion is thereby identified as it evolves in an MD simulation trajectory of the GPCR. This capability is due to the fact that the residue pair contact is inherently normalized, and thus, by following the cumulative change in the well-defined local CVs, the event detection is made more robust to (non-collective) noise. Then, the detection of temporally coordinated changes in local elements that can be distant to one another reveals function-related changes resulting from the coordinated sequence of local rearrangements during dynamics.

#### 2.4.3. The Role of The Ligand in Transitions to Functional States

The role of the ligand in determining conformational transitions to function-related states revealed from rare events detected in the dynamics of ligand bound in the orthosteric pocket of the GPCR. Application of the RED algorithm to the dynamics of the KET ligand bound in the orthosteric site of the 5-HT_2A_R was investigated, with feature data represented by the contact map limited to interactions between the ligand and the protein. The contact data vector resulting from the smoothing operation, described in Section 2.1 and Section 2.2, was of the size (*n*, 315), where *n* is the number of frames in the trajectory for the 315 residues in the protein.

Only three components needed to be considered for this smaller data vector, and additional constraints were added to increase the sparseness and temporal smoothness of the components. The weights of the three NMF components plotted against time in Figure 9A show component 2 to contain the largest absolute weights. The top 10 features (residues) with the highest weights of contribution to component 2 are highlighted in red on the 5-HT_2A_R structure in Figure 9B. For clarity, Figure 9C shows component 2 in isolation from the other two components. The event occurring during this largest spike was analyzed by observing frames before, during, and after the event at times indicated by the colored vertical lines in Figure 9D. The structures from trajectory snapshots at the times indicated by these vertical lines are shown in Figure 9F–H. The sequence of snapshots shows that component 2 represents a reorientation of the headgroup of the ligand, from pointing towards TM7 to pointing towards TM2. This is evidenced by the formation of a contact between the ligand and Ser2.61 and Thr2.64, two residues whose weight of contribution to component 2 is in the top ten. The temporal proximity of this rare event to the conformational changes in the 5-HT_2A_R discussed in previous sections is evidenced in Figure 9E, in which the largest spike, which starts around frame 820, occurs at the same trajectory time as the base of the main hump of component 2 identified in the RED analysis of the protein (not the ligand) in the 5-HT_2A_R/KET trajectory (see Figure 6B). As presented in Section 2.4.1 and Section 2.4.2 this signals the start the conformational events in the intracellular side of the receptor at the same time. This temporal sequence is thus suggestive of a causative relation between these sets of events in protein regions far from one another in the molecular scale. Whether the conformational transition observed in the protein is triggered allosterically by the reorientation of the ligand, and how, is the topic of a forthcoming analysis with the quantitative tools developed specifically for this purpose [10,18,23,24].

## 3. Discussion

The GPCRs are prototypical allosteric proteins that connect the cell environment to physiological responses that modulate the state of the cell. The molecular processes underlying the various elements of such complex signal transduction [25,26,27,28,29,30,31] include direct coupling to intracellular effector proteins in various cascades, anchoring to scaffolding proteins such as PDZ domains, as well as GPCR- and function-specific spatial organization such a dimerization and oligomerization. Regulatory effects of these processes involve structural modifications such as phosphorylation by specific classes of kinases, and interactions with the internalization machinery. The common denominators of these processes are the molecular conformational changes that transfer the GPCR from state to state to enable the interactions with the environment and functional protein partners (e.g., [9,10,11,30,31,32]). The drugs targeting GPCRs are, therefore, required and designed to intervene in this complex array of interrelated mechanisms, mostly by inducing and modulating the structural and dynamic states of the GPCR molecule. Not surprisingly, therefore, structural and computational approaches that discover and quantitatively illuminate the details of GPCR molecule responses to ligands and external stimuli are major factors in the ability to consider, understand, and perform ligand discovery and design [2,3,4]. Central among these tools and approaches are MD simulations that offer the ability to follow the dynamic changes in GPCR molecular structure in response to the environmental condition (especially the membrane), interactions with other proteins, and the effects of ligand binding [32]. The need for, and the success of, using MD simulations for to study such processes involving the allosteric mechanisms of molecular machines in the membrane are well known [32,33,34,35], and so are the challenges presented by the very large amounts of resulting data that encode the desired information [7,36,37]. This difficulty is compounded by the fact that mechanistically important state-to-state transitions of complex molecular machines such as GPCRs must constitute rare events in the MD trajectories, reflecting the need to preserve the specificity of their regulation. We have, therefore, focused on improving the ability to detect regulated allosteric mechanisms in GPCRs and other molecular machines in the membrane, and have developed the RED protocol described here for the efficient detection of rare events of conformational transitions.

The results from the application of the new RED protocol to the MD trajectories of ligand-bound 5-HT_2A_R, shown here, reveal clearly defined, structurally coherent, and mechanistically relevant events of conformational changes. The structural interpretation provided by the RED protocol further showed that these detected events are complex rearrangements of structural motifs that appear together in the same discrete time frame along the trajectory, and lead to functionally relevant states of the ligand bound GPCR. We selected the pharmacologically well characterized ligands, 5HT and Ketanserin, for the study of ligand-dependent conformational changes in the 5-HT_2A_R in order to enable a direct comparative analysis of results from the RED protocol in the functional context of activation by a full agonist, and “deactivation” by an inverse agonist. Thus, the structural expression of ligand-dependent activation and deactivation of the receptor must relate to those observed from structural determinations occurring when the binding of a GPCR agonist near its extracellular side triggers conformational rearrangements of the protein molecule to enable the binding and activation of an effector protein (e.g., a heterotrimeric G-protein, or Arrestin) at the intracellular end (e.g., [9,25,26]-and references therein). This is achieved through an allosteric process that involves multiple rearrangements of specific structural microdomains throughout the receptor [38] and leads to the measurable effects of ligands.

In accordance with this mechanistic insight, we showed here that the application of the RED protocol to the analysis of MD trajectories detected rare events of conformational transitions that led to opposing effects on the intracellular region of the 5-HT_2A_R, reflecting the opposite pharmacological properties of the ligands as an activator (agonist, 5-HT) and inactivator (inverse agonist, KET). The temporal information provided by the NMF algorithm about the events in both GPCR-ligand complexes showed them to consist of temporally coincident and coordinated changes in several structural motifs. However, while the events detected in the 5-HT_2A_R with the full agonist constitute a transition to states in which the accessible volume increased in the region of the GPCR used to couple with the G protein, the inverse agonist (KET) led to a closure of the intracellular cavity and reduction in its volume. The molecular details of the transitions toward the structurally different states emerged from the analytical power of the NMF algorithm in the RED protocol to identify the time-resolved increase in the weights of the structural components dominating the structure at different time points. For example, the roles of structural components of the motions of the ICL2 in the structural transition of the intracellular cavity of the 5-HT_2A_R/KET complex from a large volume that accommodates effector binding (e.g., [20,39]), to a small and restrictive one that does not support such an interaction, was identified in detail. This change in volume was quantified by the number of water molecules the new cavity can contain. As described in the Results section, all the specific rare events of conformational transitions detected with the RED protocol and interpreted from the structural context it provides are complex rearrangements of structural motifs that appear together in the same discrete timeframe along the trajectory. Remarkably, when the same RED protocol was applied to detect rare events in the dynamics of the KET ligand bound in the orthosteric pocket of the 5-HT_2A_R, it directly highlighted the involvement of the ligand in the functional rearrangement by detecting the reorientation of the ligand and the formation of new interactions in the binding pocket which coincided with the timing of the conformational transitions responsible for the reduction in volume of the intracellular cavity. Thus, the detection of these events was shown to be directly translatable into a structural context of ligand-specific transitions that can then be associated with specific functions of the GPCR. Our hypothesis that this mechanism is a reflection of the involvement of the ligand in an allosteric pathway is now being investigated and quantified with approaches [18,24] designed to outline the mechanistic path connecting the structurally distant sites which were shown here to exhibit temporally coordinated dynamics.

We note that this quantitative information about the ligand-specific transitions between functionally relevant states, produced by pharmacologically defined ligands, complements the demonstrated ability of another machine-learning analysis of patterns in the dynamics of the GPCR-ligand complex [7]. The algorithm identifies specific dynamic patterns in the MD trajectories of ligand-GPCR complexes to classify the bound ligands by their pharmacological properties of agonist, partial agonist, etc. Together with the RED protocol analysis, the two novel machine learning approaches offer deep insights into ligand-determined functional properties of GPCR-ligand complexes. These open a new and rigorously reproducible path towards a mechanistic understanding of these important molecular machines, and for their application in drug discovery.

## 4. Materials and Methods

Details of the procedures for building the homology model, docking the ligands, obtaining parameters, and running the simulations have been described previously [7]. The main steps are summarized briefly below.

### 4.1. Homology Model of the 5-HT_2A_R

MODELLER (v.9.18) [40] was used to generate the sets of homology models of the human 5-HT_2A_R. Three sets of structure-based sequence alignments were used as templates in Modeller:Set 1:consisted of two structures of the human 5-HT_2B_R (PDBID: 4ib4 and 5tvn);Set 2:included two structures of the human 5-HT_2B_R (PDBID: 4ib4 and 5tvn) and two structures of the human 5-HT_1B_R (PDBID: 4iaq and 4iar);Set 3:included all the structures in Set 2, augmented by 2 structures of the human β2-adrenergic receptor b2AR (PDBID: 4lde and 4ldl).

Each of these template structures includes the receptor, bound to one of its agonists. For each template set, Modeller was used to generate 1000 homology models of the 5-HT_2A_R.

Each model’s ability to discriminate between true agonist and decoy structures was quantified as described previously [7]. The set that only used 5-HT_2B_ as input templates was found to be the most discriminating model overall and was used for subsequent docking studies. 

### 4.2. Parametrization and Docking of the Molecular Models

MOL2 files for the 5-HT_2A_R ligands 5-HT and Ketanserin were obtained from the ZINC database [41] and docked into the receptor using the Induced Fit Protocol [42] in the Schrodinger Suite. A starting binding pose was chosen for each ligand based on the emodal score and comparison to experimental data presented in [7]. As shown in Appendix A, the structures of the 5-HT_2A_R/5-HT and 5-HT_2A_R/KET complexes were very similar (0.518 Å backbone RMSD).

### 4.3. Comparison of Modeled Starting Structures to 5-HT_2A_R Structures in the PDB

#### 4.3.1. Comparison of Starting Structures

After the MD trajectory data described and used in this paper were collected, structures of the 5-HT_2A_R in various states were published, including an inactive 5-HT_2A_R/risperidone complex (pdb 6A93) [21], as well as an “active” structure of a 5-HT_2A_R bound to the partial agonist 25-CN-NBOH and in complex with an engineered G_q_ (pdb 6WHA) [20]. Comparison of our starting structures to these active and inactive structures reveals that our starting structures are in a “partially active” conformation.

The RMSD of the transmembrane domains between the inactive structure and the superposed active state structure is 1.9 Å. The RMSDs of the transmembrane domains between the inactive structure and the superposed Ketanserin-bound and 5-HT-bound homology model starting structures are 1.4 and 1.6 Å, respectively. Transmembrane domains were defined as: TM1-Val1.42 to Val1.57; TM2-Thr2.39 to Iso2.25; TM3 –Cys3.25 to Val3.52; TM4–Ala4.42 to Phe4.63; TM5–Asn5.37 to Lys5.63; TM6–Glu6.31 to Met6.57; TM7–Iso7.31 to Val7.52 (Ballesteros and Weinstein generic numbering systems defined in [17])

For the intracellular side of the transmembrane domains, the RMSD between the inactive structure and the superposed active state structure is 2.2 Å. The RMSDs of the intracellular side of the transmembrane domains between the inactive structure, and the superposed starting structures of the 5-HT_2A_R/Ketanserin and 5-HT_2A_R/5-HT homology model, are 1.56 and 1.56 Å, respectively. The intracellular ends of transmembrane domains were defined as: TM1-Iso1.47 to Val1.57; TM2-Thr2.39 to Leu2.52; TM3–Ser3.39 to Val3.52; TM4–Ala4.42 to Thr4.51; TM5–Leu5.51 to Lys5.63; TM6–Glu6.31 to Leu6.43; TM7–Ser7.45 to Val7.52

#### 4.3.2. Structures Resulting from the MD Simulations: 5-HT_2A_R/KET vs. 5-HT_2A_R/Risperidone Binding Mode

Appendix A shows the binding mode of our 5-HT_2A_R/KET starting structure (green) superposed with the pdb 6A93 structure of the 5-HT_2A_R/risperidone complex (lavender). Our predicted binding mode of Ketanserin is very similar to the binding mode of risperidone. One difference is that the extracellularly oriented headgroup of Ketanserin is oriented towards TM2, while risperidone’s is oriented towards TM7. It is worth noting that, during the simulation, Ketanserin samples conformations in which the headgroup reorients towards TM7 (similar to the risperidone structure), and that this is one of the rare events picked up by the RED protocol in the “deactivation” mechanism (see Figure 6).

#### 4.3.3. Comparison of Functional Motifs, “Toggle Switch” W6.48

Appendix A shows a comparison of the orientation of the W6.48 in the “toggle switch” for the 5-HT_2A_R/risperidone (blue), 5-HT_2A_R/G_q_ (violet), 5-HT_2A_R/KET (green), and 5-HT_2A_R/5-HT (coral) structures. In 5-HT_2A_R/risperidone, W6.48 is flipped “up” towards the extracellular end of the receptor. In 5-HT_2A_R/G_q_, W6.48 is flipped “down” towards the intracellular side. In the 5-HT_2A_R/KET and 5-HT_2A_R/5-HT starting structures, W6.48 is flipped “up” but is shifted such that it lies in between the inactive and active state structures.

#### 4.3.4. Comparison of Functional Motifs, Intracellular Orientation of TM6

Appendix A shows a comparison of the position of the intracellular end of TM6 for the 5-HT_2A_R/risperidone (blue), 5-HT_2A_R/G_q_ (violet), 5-HT_2A_R/KET (green), and 5-HT_2A_R/5-HT (coral) structures. In 5-HT_2A_R/risperidone, TM6 orients inwards, occluding the intracellular cavity where the effector protein binds in active receptors. In 5-HT_2A_R/G_q_, TM6 is kicked “outwards” in order to accommodate the G-protein. In the 5-HT_2A_R/KET and 5-HT_2A_R/5-HT starting structures, TM6 is located in an orientation intermediate between the active and inactive structures.

#### 4.3.5. Comparison of Functional Motifs, Intracellular Orientation of TM7

Appendix A shows a comparison of the position of the intracellular end of TM7 for the 5-HT_2A_R/risperidone (blue), 5-HT_2A_R/G_q_ (violet), 5-HT_2A_R/KET (green), and 5-HT_2A_R/5-HT (coral) structures. In 5-HT_2A_R/risperidone, TM7 orients outwards, away from the center of the receptor. In 5-HT_2A_R/G_q_, TM7 is located inwards, towards the center of the receptor. In the 5-HT_2A_R/KET and 5-HT_2A_R/5-HT starting structures, TM7 is inwards, similar to the active structure.

#### 4.3.6. Comparison of Functional Motifs, the Ionic Lock

Appendix A shows a comparison of the position of the ionic lock (R3.50-E6.30) between TM3 and TM6 for the 5-HT_2A_R/risperidone (blue), 5-HT_2A_R/G_q_ (violet), 5-HT_2A_R/KET (green), and 5-HT_2A_R/5-HT (coral) structures. In 5-HT_2A_R/risperidone (Appendix A), the ionic lock is formed, as expected for an inactive conformation. In 5-HT_2A_R/G_q_ (Appendix A), the ionic lock is broken, as expected for an active conformation. In the 5-HT_2A_R/KET and 5-HT_2A_R/5-HT starting structures (Appendix A), the ionic lock is broken, but closer together than in the active state structure.

### 4.4. Molecular Dynamics Simulations

The 5-HT_2A_R-ligand complexes were inserted into a membrane containing 144:16 POPC:Cholesterol molecules of in each leaflet using the CHARMM-gui [43]. Each complete system was equilibrated under the NPT ensemble (T = 310 K) in NAMD according to a previously established multistep equilibration protocol (see [7] for details). The final frames of the equilibration were used as an input to run MD simulations of the systems under the NVT ensemble (T = 310 K) using the OpenMM software [44]. The simulations were run as 6 replicas for each system. The analysis was performed on ~3-microsecond-long trajectories for each of the complexes.

### 4.5. Calculation of the Intracellular Cavity Volume

The volume of the intracellular cavity was defined based on a procedure used to quantify the volume of vestibules in another membrane protein [45]. Briefly, a water molecule is considered to be in the intracellular cavity if its oxygen atom is within 15 Å of the C-alpha atom of L2.43, or within 8 Å of the C-alpha atom of A6.33, but not within 5 Å of lipid atoms, and not within 12 Å of the C-alpha atom of S3.39.

### 4.6. Visualization of Intracellular Cavity Volume

PDBs corresponding to the frames in the analysis of the events during which the volume changed (frames 420, 510, and 620 for serotonin-5-HT_2A_R complex trajectory, see Figure 1E–G) and (frames 617, 730, 770, 970, and 1197 for Ketanserin-5-HT_2A_R complex trajectory, see Figure 6C–G) were used as input to the CASTp webserver [46] automated volume calculator, using the default probe radius of 1.4Å.

## Figures and Tables

**Figure 1 molecules-26-03059-f001:**
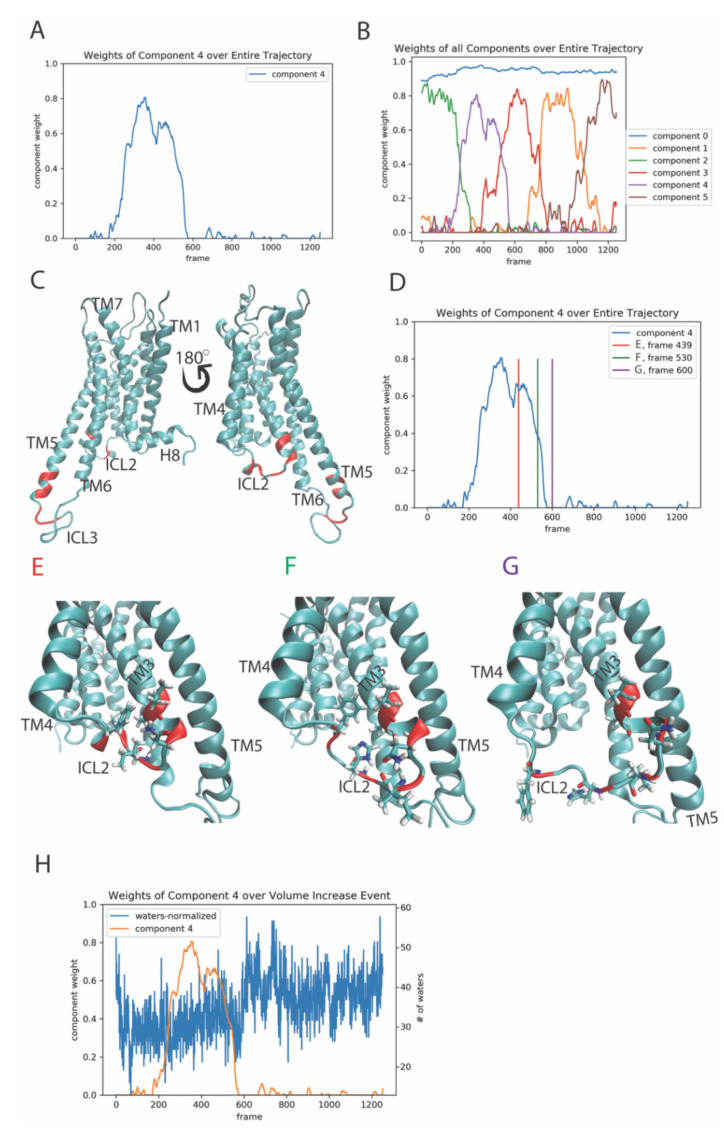
Panel (**A**)-Evolution of the weights of the 4th output component of the NMF decomposition of the smoothened contact data calculated from the trajectory of the 5-HT_2A_R complex with 5-HT, plotted over the frames of the trajectory. Panel (**B**)–Evolution of the weights of all 6 components plotted over the frames of the trajectory. Panel (**C**)-Structure of the receptor showing (in red) the top ten features (residue pairs) contributing to component 4. The residues are identified with the generic numbering system for GPCRs [17].: Val3.52, Ala3.53, Asn3.56, Iso3.58, His3.60, Phe3.63, Val5.73, Ser5.74, Leu5.76, Gly5.77, Ala5.80, Lys5.81, Leu5.82. Panel (**D**)-The colored vertical lines on the weight profile of component 4 denote the trajectory frames of the GPCR/ligand complex shown in Panels (**E–G**)**,** representing the structure before, during and after the event dominated by component 4. Panels (**E–G**) Snapshots of the receptor at the frames indicated by the colored lines in Panel (**G**). The Intracellular Loop 2 (ICL2) residues that are among the top ten features (residue pairs) contributing to component 4, are highlighted in red and shown in stick model. Panel (**H**) Component 4 (orange line) plotted alongside the number of water molecules in the intracellular cavity of the receptor over the frames of the trajectory (see methods for details). The scalebar on the right side of the graph represents the number of waters in the cavity, while the left scalebar is normalized to 1 for comparison with component 4.

**Figure 2 molecules-26-03059-f002:**
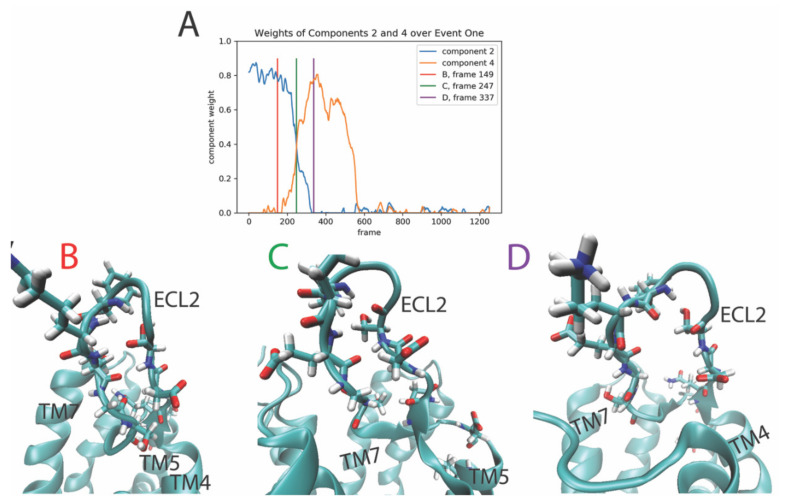
Panel (**A**)–Evolution of the weights of the 2nd and 4th output components of the NMF decomposition of the smoothened contact data calculated from the trajectory of the 5-HT_2A_R-serotonin complex, plotted over the frames of the trajectory. Panels (**B**–**D**)–Snapshots of the receptor at the frames indicated by the colored lines in Panel (**A**). The ECL2 residues that are among the top ten features (residue pairs) contributing to component 2 and 4, are shown in stick model.

**Figure 3 molecules-26-03059-f003:**
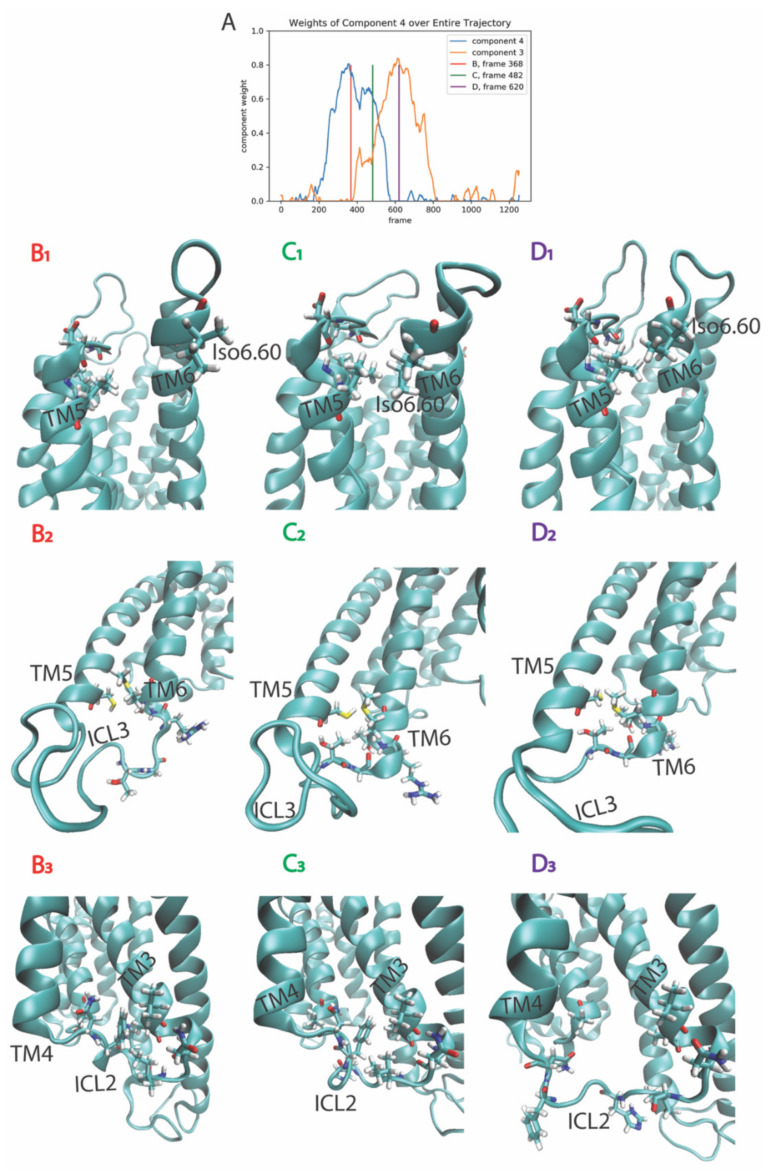
Panel (**A**) Evolution of the weights of the 3rd and 4th output components of the NMF decomposition of the smoothened contact data calculated from the trajectory of the 5-HT_2A_R-serotonin complex, plotted over the frames of the trajectory. Panels (**B_1_**–**D_1_**) Snapshots of the receptor at the frames indicated by the colored lines in Panel (**A**), zoomed to show the rearrangement of the extracellular ends of TM5 and TM6. The residues shown in stick model (Iso6.60, Leu5.40, Val5.39, and Asp5.36) are identified by the RED as among the top ten features (residue pairs) contributing to component 3 and 4. Panels (**B_2_**–**D_2_**) Snapshots of the receptor at the frames indicated by the colored lines in Panel (**A**), zoomed to show the rearrangement of the intracellular end of TM6 (addition of a helical turn). The residues shown in stick model (Cys5.72, Thr6.19, Gly6.20, Arg6.22, Thr6.23, and Met6.24) are identified by the RED as among the top ten features (residue pairs) contributing to component 3 and 4. Panels (**B_3_**–**D_3_**) Snapshots of the receptor at the frames indicated by the colored lines in Panel (**A**), zoomed to show the rearrangement of the ICL2 from alpha helical to unstructured. The residues shown in stick model (Ala2.38, Val3.52, Ala3.53, Asn3.56, Iso3.58, His3.60, Phe3.63, and Asn4.37) are identified by the RED as among the top ten features (residue pairs) contributing to components 3 and 4.

**Figure 4 molecules-26-03059-f004:**
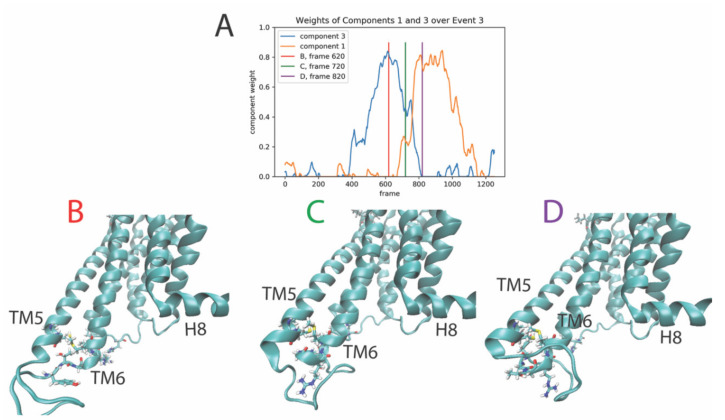
Panel (**A**) Evolution of the weights of the 1st and 3rd output components of the NMF decomposition of the smoothened contact data calculated from the trajectory of the 5-HT_2A_R-serotonin complex, plotted over the frames of the trajectory. Panels (**B**–**D**) Snapshots of the receptor at the frames indicated by the colored lines in Panel (**A**), zoomed to show the rearrangement of the intracellular end of TM6. The residues shown in stick model (Cys5.72, Thr6.19, Gly6.20, Arg6.22, Thr6.23, Met6.24, Leu5.71, Tyr6.18, Pro3.57, and His3.60) are identified by the RED as among the top ten features (residue pairs) contributing to component 3 and 1.

**Figure 5 molecules-26-03059-f005:**
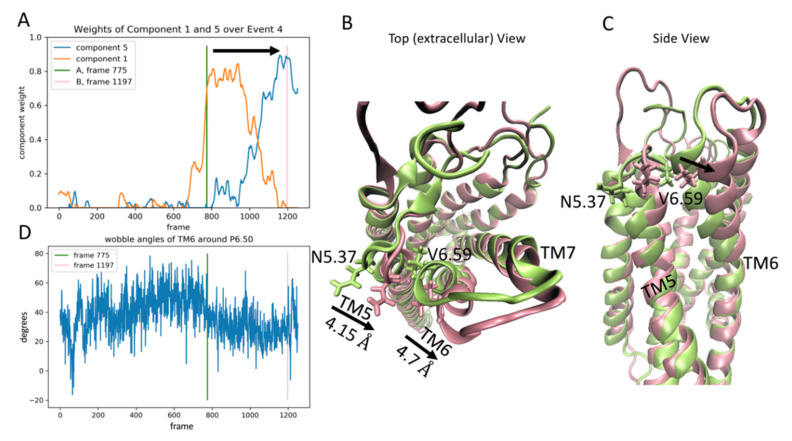
Panel (**A**) Evolution of the weights of the 1st and 5th output components of the NMF decomposition of the smoothened contact data calculated from the trajectory of the 5-HT_2A_R-serotonin complex, plotted over the frames of the trajectory. Panels (**B**,**C**) Snapshots of the receptor at the frames indicated by the colored lines in Panel (**A**) (green structure at frame 775 and pink structure at frame 1197). The residues N5.37 and V6.59 are shown in stick model and colored according to frame (green for frame 775 and pink for frame 1197). Panel (**D**) Evolution of the wobble angle of the proline kink around 6.50 (see Section 2.3.4 for details) calculated from the trajectory of the 5-HT_2A_R-serotonin complex, plotted over the frames of the trajectory. The green and pink lines correspond to the same frames shown by the colored lines of panel A and the colored structures in panels B-C (green for frame 775 and pink for frame 1197).

**Figure 6 molecules-26-03059-f006:**
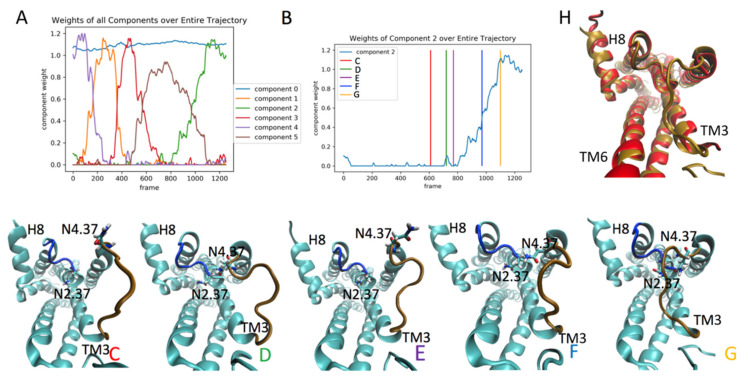
Panel (**A**) shows the evolution of the weights of all 6 components from the RED analysis of the Ketanserin (KET)-bound 5-HT_2A_R. Panel (**B**) shows the weight profile of component 2. The colored vertical lines denote the trajectory frames corresponding to the snapshots of the GPCR/KET complex shown in panels (**C**–**G**), which represent the structure before, during and after the event, dominated by component 2. The 5-HT_2A_R-Ketanserin complex is show in cyan, with the ICL1 colored in blue, the ICL2 colored in brown, and N4.37 and N2.37 shown in stick model. Panel (**H**) shows a superposition of the crystal structure of the 5-HT_2A_R bound to the inverse agonist risperidone (pdb 6A93, in red) with the structure of the 5-HT_2A_R-Ketanserin complex (colored gold) at the time corresponding to frame G (gold line in panel (**B**).

**Figure 7 molecules-26-03059-f007:**
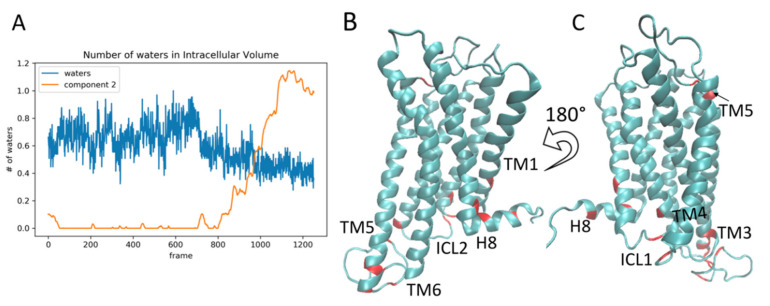
Panel (**A**) Component 2 (orange line) plotted alongside the number of water molecules in the intracellular cavity of the receptor over the frames of the trajectory (see methods for details). The scalebar is normalized to 1 for comparison with component 2. Panels (**B**,**C**) show snapshots of the receptor, with the residues constituting the top ten contributing residue pairs of component 2, highlighted in red. These residues are Leu1.52, Asn2.37 (ICL1), Tyr2.41, Glu3.55, Asn4.37 (ICL2), Asp5.35, Val5.39, Leu5.71, Cys5.72, Asp5.75, Leu5.76, Arg5.79(ICL3), Ser5.84 (ICL3), Arg6.21 (ICL3), Leu7.55, Arg7.61 (Helix 8), Tyr7.67 (Helix 8).

**Figure 8 molecules-26-03059-f008:**
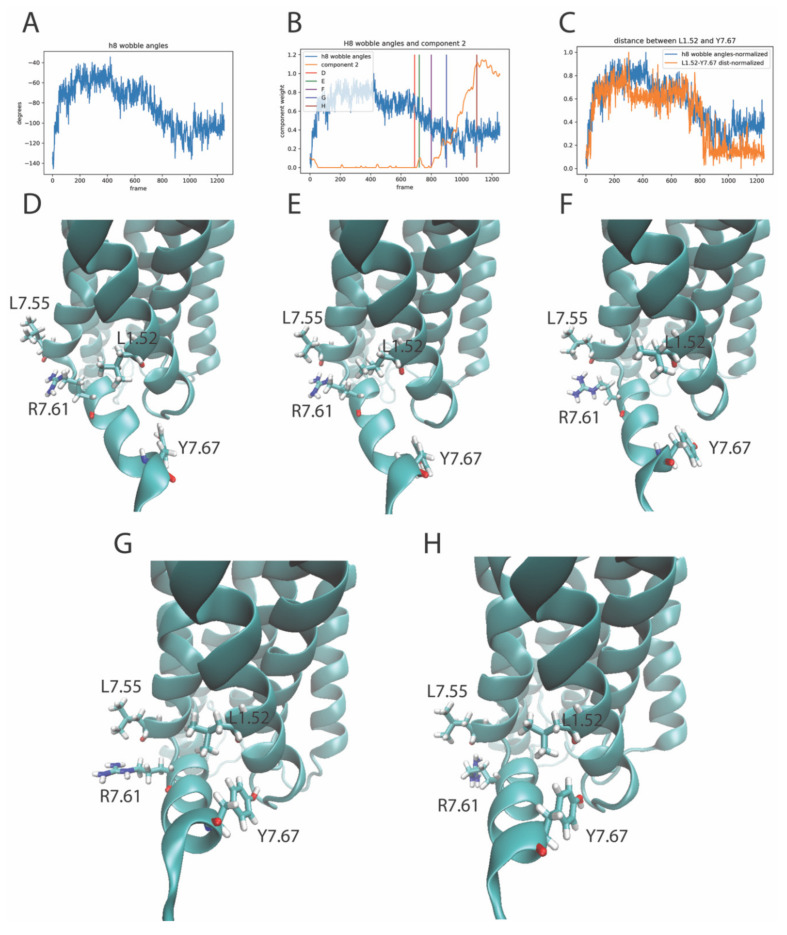
Panel (**A**) Evolution of the wobble angle of Helix 8 (see Section 2.4.2 for details) calculated from the trajectory of the 5-HT_2A_R-ketanserin complex, plotted over the frames of the trajectory. Panel (**B**)-Component 2 (orange line) plotted alongside the Helix 8 wobble angle over the frames of the trajectory. The scalebar is normalized to 1 for comparison with component 2. Colored vertical lines denote the trajectory frames of the GPCR/ligand complex shown in Panels (**D**–**H**), representing the structure before, during and after the event, dominated by component 2. Panel (**C**) Helix 8 wobble angles (blue line) plotted alongside the distance between the alpha carbons of L1.52-Y7.67 (orange line) over the frames of the trajectory. Panels (**D**–**H**) Snapshots of the receptor at the frames indicated by the colored lines in Panel (**B**). The residues shown in stick model (L7.55, R7.61, L1.52, and Y7.67) are among the top ten features (residue pairs) contributing to component 2.

**Figure 9 molecules-26-03059-f009:**
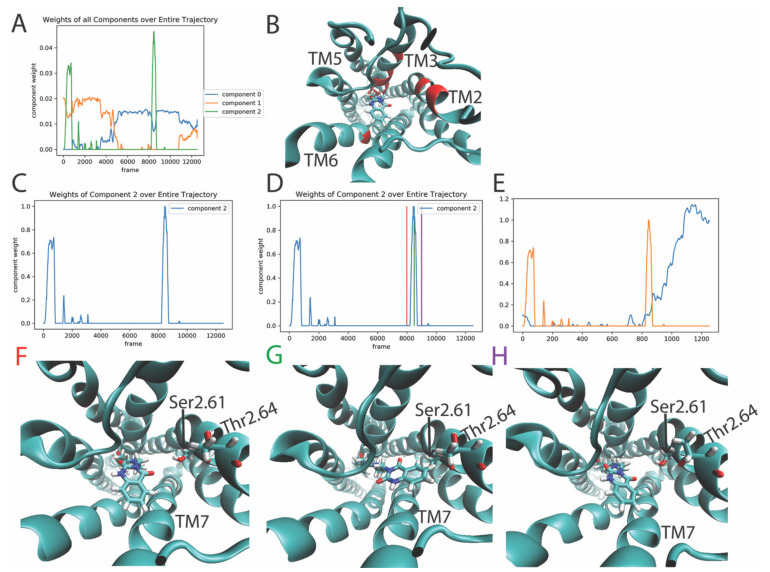
Panel (**A**) Evolution the weights of all 3 components of the NMF decomposition of the smoothened ligand contact data (see Section 2.4.3 for details) calculated from the trajectory of the 5-HT_2A_R-ketanserin complex, plotted over the frames of the trajectory. Panel (**B**) Structure of the receptor showing (in red) the top ten features (residue pairs) contributing to component 2. The residues are Ser2.61, Thr2.64, Trp3.28, Iso2.65, Val3.33, Thr3.37, Phe6.51, D3.32, Iso3.40, and Ser3.36. Panel (**C**) Evolution of the weights of the 2nd output component. Panel (**D**) Evolution of the weights of the 2nd output component with colored vertical lines on the weight profile of component 2 denoting the trajectory frames of the GPCR/ligand complex shown in Panels (**F**–**H**), representing the structure before, during and after the event dominated by component 2. Panel (**E**) Evolution of the weights of the 2nd output component of NMF decomposition of the smoothened ligand contact data (orange line, see Section 2.4.3), plotted alongside the 2nd output component of NMF decomposition of the smoothened contact data of the receptor (blue line, see Section 2.4.1) Panels (**F**–**H**) Snapshots of the receptor at the frames indicated by the colored lines in Panel (**D**). Ser2.61 and Thr2.62, shown in stick model, are among the top ten features (residue pairs) contributing to this component.

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
