# Peer review of "Ligand-Dependent Conformational Transitions in Molecular Dynamics Trajectories of GPCRs Revealed by a New Machine Learning Rare Event Detection Protocol"

_molecules, 2021, doi:10.3390/molecules26103059_

Round 1

Reviewer 1 Report

I think this is a great study, well implemented, and an important step in the understanding and description of ligand-determined functional mechanisms of GPCRs.

The background is exhaustively an clearly described in the introduction and the open questions the work addresses are clearly stated.  The potentiality of the unsupervised non-negative matrix factorization machine learning technique used are wisely exploited. The analysis of the results is accurate and the conclusions are sound.

Therefore the paper can be accepted as it is for publication on the special issue "G Protein-Coupled Receptors and Transporters in the CNS as Drug Targets"

Author Response

We are grateful to this Reviewer for the positive feedback and acceptance of our work.

Reviewer 2 Report

In this manuscript, a new machine learning Rare Event Detection (RED) protocol was developed to investigate ligand-dependent conformational transitions in molecular dynamics (MD) simulations of GPCRs. The RED protocol is based on an unsupervised machine learning technique called non-negative matrix factorization (NMF) for analysis of sparse, non-negative data obtained from the MD trajectories. As demonstrated on the serotonin 5-HT2AR receptor, the RED analysis was able to identify distinct conformational transitions of the GPCR in response to binding of pharmacologically different ligands, including the serotonin (5-HT) agonist and Ketanserin (KET) inverse agonist. The analysis nicely related the output components to archetypes of protein conformational changes, weights to contributions of each feature (residue pair), and importantly interlocking/overlapping humps of the components to rare transition events. However, a number of suggestions that could potentially help improving the manuscript include:

1. The presented analysis used the NMF algorithm with 6 components. How is the number of components chosen? How do we make sure this number of components is able to capture all important conformational transitions between different states of the systems? Are we expecting similar RED analysis results if using a different number of components?

More generally, given a new GPCR simulation, are there certain criteria that can be used to help us deciding how many components to analyze? 

2. Only one single MD trajectory was used for separate RED analysis of the two simulation systems, i.e., 5-HT and KET-bound receptor. The authors noted that "Interestingly, component 2 in the receptor complex with the inverse agonist KET, corresponds to component 4 in the agonist-bound 5-HT2AR complex." In this case, if MD trajectories of the two systems were combined for RED analysis, would it give the same component in the corresponding time windows?

On the other hand, can the authors test if similar components are obtained if each current MD trajectory is divided into two halves? How are the components compared between two subsequent time window, and with those from the entire trajectory?  

3. Related to the above point, this could be related to simulation convergence issues. But multiple independent MD simulation are generally preferred to improve conformational sampling, especially for complex biomolecules such as GPCRs. If RED is applied to such multiple simulations, would it generate the similar components and similar rare events of conformational transitions in the system?

4. For reproducibility, what software did the authors use to build the contact maps? What specific atoms were used to calculate the distance (< 3.5Å for being in contact between two residues)?

5. Figure 1 on page 24 should be "Figure A1"?

6. Lines 182-184: Clarification is needed for sentences "The occurrence of a new conformational transition event is identified by an overlap in time (i.e., occurring at the same trajectory frames) of the decreasing and increasing weights of two consecutive components." 
Related, "The first event is defined by the transition of dominance from component 2 to component 4." Apparently, 2 and 4 are not consecutive, similarly for third event (lines 315-317) and fourth event (lines 333-334).
It seems more accurate to say "dominant components in two consecutive time windows".

Author Response

Please see attached file. Thank you

In this manuscript, a new machine learning Rare Event Detection (RED) protocol was developed to investigate ligand-dependent conformational transitions in molecular dynamics (MD) simulations of GPCRs. The RED protocol is based on an unsupervised machine learning technique called non-negative matrix factorization (NMF) for analysis of sparse, non-negative data obtained from the MD trajectories. As demonstrated on the serotonin 5-HT2AR receptor, the RED analysis was able to identify distinct conformational transitions of the GPCR in response to binding of pharmacologically different ligands, including the serotonin (5-HT) agonist and Ketanserin (KET) inverse agonist. The analysis nicely related the output components to archetypes of protein conformational changes, weights to contributions of each feature (residue pair), and importantly interlocking/overlapping humps of the components to rare transition events. However, a number of suggestions that could potentially help improving the manuscript include:

We thank the reviewer of this summary and for the intention to help improve the manuscript.

  1. The presented analysis used the NMF algorithm with 6 components. How is the number of components chosen? How do we make sure this number of components is able to capture all important conformational transitions between different states of the systems? Are we expecting similar RED analysis results if using a different number of components?

We are delighted by the Reviewer’s interest and glad to answer the series of questions in some detail here and briefly in the Supplementary Information. We considered the choice of the factorization rank for the results described in the manuscript to be a technical detail that would take away from the focus on the observation of ligand-determined conformational changes, because the topic requires a discussion at a higher level than seemed appropriate for this manuscript that describes the principles of the method and its use in analyzing the structural dynamics of GPCR function. For this reason, the detailed response below is not included in the revision of the manuscript.

The NMF method belongs to the category of unsupervised learning tasks for which the optimal choice of rank (i.e., number of components) is user-defined based essentially on prior knowledge of the system. This is akin to the choice of collective variables to represent the motions of a simulated system protein as used in dimensionality reduction of MD trajectory data by projection into spaces such as tICA. Indeed, in a 2014 review article, Nicolas Gillis proposes that the three main ways to pick the proper rank of NMF are 1) expert insight, 2) trial and error, and 3) SVD (see expert below), saying that “The choice of the factorization rank r, that is, the problem of order model selection, is usually rather tricky. Several popular approaches are: trial and error (that is, try different values of r and pick the one performing best for the application at hand), estimation using the SVD (that is, look at the decay of the singular values of the input data matrix), and the use of experts insights (e.g., in blind HU, experts might have a good guess for the number of endmembers present in a scene); see also [7, 104, 70] and the references therein”

-- (Gillis, N. "The why and how of nonnegative matrix factorization, 2014.") – we will add this description and reference to the manuscript, and thank the Reviewer for the suggestion.

We illustrate briefly here the iterative control/evaluation of the selected ranking for the problem discussed in the manuscript. The results from NMF analysis of the 5-HT2AR bound to serotonin using 3 components to 8 components are summarized in figure A2, below. With 3 components (figure A2, top left) there is no steady state solution. With 4 components (figure A2, top middle) there is a steady state solution, which is congruent with our expert knowledge that there will be a subset of the receptor contacts that do not change (i.e. backbone hydrogen bonds between stable parts of the transmembrane helices). With 5 components (figure A2, top right) the steady state solution, as well as the two events detected by the analysis with 4 components, remain extant, but there is an additional event detected around frame 450 (~1microsecond). This event is ~500ns away from the other events and seems congruent with being a separate slow (rare) event. Thus we accept the 5 component NMF as being better than the 4 component NMF. Similarly, 6 component NMF (figure A2, bottom left) adds another rare event around frame 1050 (~2.5 us) and we conclude that it is better than 5 component NMF. NMF with 7 components indeed adds a new event around frame 70 (~168 ns), however, this is obviously too early in the trajectory of such a complex system to be considered anything but related to the relaxation of the MD simulation, rather than to the function of the protein, and thus the 7 component NMF does not add new functional information. The NMF with 8 components (figure A2, bottom right) bifurcates into two events the single event which was around frame 750 in the 6 component NMF (figure A2, bottom left). These bifurcated events are close together and are thus likely to be part of the same slow motion (allosterically connected). While the user could choose to analyze these bifurcated events as two separate contemporaneous motions, for the analysis of allosteric mechanisms it is more informative to consider them to be part of the same event (this also reduces model complexity and makes analysis more efficient for the user). Adding more components continues to bifurcate the slow events into multiple faster ones. We thus concluded that 6 components is an appropriate choice for the analysis of this trajectory as it captures all of the detectable functional motions with the least amount of model complexity.

More generally, given a new GPCR simulation, are there certain criteria that can be used to help us deciding how many components to analyze?

      Clearly, the method can be applied to any new/other GPCR trajectory. A good starting point for estimating the number of components is to consider the expected number of rare events in the trajectory based on mechanistic hypotheses about conformational changes required for the molecular process. The user’s expert knowledge of the system provides an advantage, but as the analysis can be readily redone with +/- (2-3) components, convergence can be readily attained. This information was added to the revised manuscript in lines 167-178.

Figure A2. NMF of the trajectory of the 5-HT2AR bound to serotonin, computed with 3 components (top left), to 8 components (bottom right).

  1. Only one single MD trajectory was used for separate RED analysis of the two simulation systems, i.e., 5-HT and KET-bound receptor. The authors noted that "Interestingly, component 2 in the receptor complex with the inverse agonist KET, corresponds to component 4 in the agonist-bound 5-HT2AR complex." In this case, if MD trajectories of the two systems were combined for RED analysis, would it give the same component in the corresponding time windows?

We are concerned that this question reflects a likely misunderstanding of the NMF analysis and its results.

First, we note that the MD trajectory data for each ligand-GPCR complex are not “a single trajectory”. The data were collected from swarms of 6 replica runs of 3 microseconds each for each GPCR complex. The analysis was applied to the 3 microseconds  trajectories as mentioned in the text. This is now clarified in lines 823-827 of the revised manuscript.

Second, it should be clear from the detailed description of the methodology (see Fig. 1) and the description results in each figure of the text, that the “components” in this application of the NMF to MD trajectory analysis are combinations of collective variables based on distances. The correspondence pointed out in the text is based on the composition of the components, not on the time of the dominance in the trajectory.

Third, there does not seem any good reason to combine trajectories of different systems. But if the two trajectories were combined (i.e. concatenated, not admixed) as suggested by the Reviewer, then each event would occur at its time in the evolution of the concatenated trajectory.

No other “combination” of trajectories of two completely different systems: one being the receptor protein bound to 5-HT, and another of the same receptor bound to KET, is possible. Therefore, the change in structural dominance will be detected when it occurs in the corresponding trajectory, without reference to what comes next or went before, at the time in the combined (concatenated trajectory) at which they occur. 

  1. On the other hand, can the authors test if similar components are obtained if each current MD trajectory is divided into two halves? How are the components compared between two subsequent time window, and with those from the entire trajectory?

It is somewhat difficult to understand the gist of question given the fact that the analysis with NMF is clearly time-resolved and the rare events it detects occur at specified times in the trajectory. If an event does not occur in a particular half of the trajectory, then it should not be detected and there is no reason to expect that it will (or would) complicate in any way the normal identification by the NMF of events occurring in the (concatenated) trajectory of the other system. Perhaps the question reflects a concern about convergence of a trajectory to a specific equilibrium state, which takes time to occur and is the usual concern about MD simulations, but this is not a valid concern in this case as the events are identified and registered as they evolve in the trajectory, and not from some sort of equilibrium average.

  1. Related to the above point, this could be related to simulation convergence issues. But multiple independent MD simulation are generally preferred to improve conformational sampling, especially for complex biomolecules such as GPCRs. If RED is applied to such multiple simulations, would it generate the similar components and similar rare events of conformational transitions in the system?

As mentioned above, this study involved the generation of 6 trajectories for each ligand system.

While not all trajectories exhibited the full set of function-related conformational transitions, they did undergo some similar conformational transitions which the RED protocol detected with components highlighting the same structural features. For example, when the RED protocol was applied to a different trajectory of the 5HT bound to the 5HT2AR, the first detected event (see Figure A3(A)) involved an “unzipping” of the secondary structure of the ECL2 similarly to the first event detected by the RED protocol when applied to the analogous trajectory described in the manuscript (compare panels in Figure A3 here to Figure 2 in the manuscript).

Figure A3. Panel A shows the results of the RED protocol application to a trajectory of 5HT bound the 5HT2AR (a separate replicate to the one presented in the main text). Panel B zooms in on the first event involving components 1 and 2. The colored lines correspond to frames before, during, and after the event. Panels C-E show the receptor structure at the times corresponding to the frames represented by the colored labeled lines in panel B. The top ten residue pairs contributing to the event are shown in stick model. These residues are: R2.70, C3.25, S3.47, L4.65, D4.68, S4.69, K4.70, F4.72, K4.73, E4.74, G4.75, C4.77, D5.75, F6.41, T7.54, and F7.64.

  1. For reproducibility, what software did the authors use to build the contact maps? What specific atoms were used to calculate the distance (< 3.5Å for being in contact between two residues)?

The contact maps were built using the atomselect command in VMD “[atomselect top "protein and same residue as within 3.5 of (protein and resid $rez)" frame $fr]” where $rez is a variable representing a particular residue and $fr is a variable representing a particular frame. This command is looped over all protein residues and frames. The VMD manual states:

“[within]…selects all atoms within the specified distance (in Å) from a selection, including the selection itself. Therefore, the command:

        within 5 of name FE

selects all atoms within 5 Å of atoms named FE.”

Thus, the distance between each pair of atoms of every residue pair is considered. If any of these distances are less than or equal to 3.5 Å, then the residues are considered to be in contact. This information has been added to the revised manuscript on lines 99-105

  1. Figure 1 on page 24 should be "Figure A1"?

Indeed. We thank the reviewer for catching this typographical error. We have changed the name to “Figure A1” in the revised manuscript.

  1. Lines 182-184: Clarification is needed for sentences "The occurrence of a new conformational transition event is identified by an overlap in time (i.e., occurring at the same trajectory frames) of the decreasing and increasing weights of two consecutive components." 
    Related, "The first event is defined by the transition of dominance from component 2 to component 4." Apparently, 2 and 4 are not consecutive, similarly for third event (lines 315-317) and fourth event (lines 333-334).
    It seems more accurate to say "dominant components in two consecutive time windows".

As shown in Figure 1B, components 2 and 4 are temporally consecutive. The names of the components are totally arbitrary for each system analyzed, and they do not determine either their sequence of appearance or their importance (i.e., the timing of their dominance in the trajectory, or the extent of the conformational change). Thus, the observation about the time of dominance of component 2 and its corresponding component 4 in the other system is just an identification of the timing of the conformational rearrangement described by the CVs in this component.
